# Non-normal Recurrent Neural Network (nnRNN): learning long time dependencies while improving expressivity with transient dynamics

**Giancarlo Kerg**[1,2,*]     **Kyle Goyette**[1,2,3,*]     **Maximilian Puelma Touzel**[1,4]

**Gauthier Gidel**[1,2]    **Eugene Vorontsov**[1,5]    **Yoshua Bengio**[1,2,6]    **Guillaume Lajoie**[1,7]

## Abstract

A recent strategy to circumvent the exploding and vanishing gradient problem in RNNs, and to allow the stable propagation of signals over long time scales, is to constrain recurrent connectivity matrices to be orthogonal or unitary. This ensures eigenvalues with unit norm and thus stable dynamics and training. However this comes at the cost of reduced expressivity due to the limited variety of orthogonal transformations. We propose a novel connectivity structure based on the Schur decomposition, though we avoid computing it explicitly, and a splitting of the Schur form into normal and non-normal parts. This allows to parametrize matrices with unit-norm eigenspectra without orthogonality constraints on eigenbases. The resulting architecture ensures access to a larger space of spectrally constrained matrices, of which orthogonal matrices are a subset. This crucial difference retains the stability advantages and training speed of orthogonal RNNs while enhancing expressivity, especially on tasks that require computations over ongoing input sequences.

## 1 Introduction

Training recurrent neural networks (RNN) to process temporal inputs over long timescales is notoriously difficult. A central factor is the exploding and vanishing gradient problem (EVGP) [13, 4, 27], which stems from the compounding effects of propagating signals over many iterates of recurrent interactions. Several approaches have been developed to mitigate this issue, including the introduction of gating mechanisms (e.g. [15, 13]), purposely using non-saturating activation functions [5], and manipulating the propagation path of gradients [3]. Another way is to constrain connectivity matrices to be orthogonal (and more generally, unitary) leading to a class of models we refer to as *orthogonal RNNs* [24, 21, 19, 33, 16, 32, 10, 2]. Orthogonal RNNs have eigen- and singular-spectra with unit norm, therefore helping to prevent exponential growth or decay in long products of Jacobians associated with EVGP. They perform exceptionally well on tasks requiring memorization of inputs over long time-scales [11] (outperforming gated networks) but struggle on tasks involving continued

1: Mila - Quebec AI Institute, Canada
2: Université de Montréal, Département d'Informatique et Recherche Opérationelle, Montreal, Canada
3: Université de Montréal, CIRRELT, Montreal, Canada
4: IVADO post-doctoral fellow
5: Ecole Polytechnique de Montréal, Montreal,Canada
6: CIFAR senior fellow
7: Université de Montréal, Département de Mathématiques et Statistiques, Montreal, Canada

Correspondence to: <lajoie@dms.umontreal.ca>

computations across timescales. A contributing factor to this limitation is the mutually orthogonal nature of connectivity eigendirections which substantially limits the space of solutions available to orthogonal RNNs.

In this paper, we propose a first step toward a solution to this expressivity problem in orthogonal RNNs by allowing non-orthogonal eigenbases while retaining control of eigenvalues' norms. We achieve this by leveraging the *Schur* decomposition of the connectivity matrix, though we avoid the need to compute this costly factorization explicitly. This provides a separation into "diagonal" and "feed-forward" parts, with their own optimization constraints. Mathematically, our contribution amounts to adding "non-normal" connectivity, and we call our novel architecture *non-normal RNN* (nnRNN). In linear algebra, a matrix is called *normal* if its eigenbasis is orthogonal, and *non-normal* if not. Orthogonal matrices are normal, with eigenvalues of norm 1 (i.e. on the unit circle). In recurrent networks, normal connectivity produces dynamics solely characterized by the eigenspectrum while non-normal connectivity allows transient expansion and compression. Transient dynamics have known computational advantages [12, 8, 9], but orthogonal RNNs cannot produce them. The added flexiblity in nnRNN allows such transients, and we show analytically how they afford additional expressivity to better encode complex inputs, while at the same time retaining efficient signal propagation to learn long-term dependencies. Through a series of numerical experiments, we show that the nnRNN provides two main advantages:

1. On tasks well suited for orthogonal RNNs, nnRNN learns orthogonal (normal) connectivity and matches state-of-the art performance while training as fast as orthogonal RNNs.

2. On tasks requiring additional expressivity, non-normal connectivity emerges from training and nnRNN outperforms orthogonal RNNs.

From a parametric standpoint, this advantage can be attributed to the fact that the nnRNN has access to *all* matrices with unit-norm eigenspectra, of which orthogonal ones are only a subset.

## 2 Background

### 2.1 Unitary RNNs and constrained optimization

First outlined in [2] and inspired by [30, 34, 18], RNNs whose recurrent connectivity is determined by an orthogonal, or unitary matrix are a direct answer to the EVGP since their eigenspectra and singular spectra exactly lie on the complex unit circle. The same mechanism was invoked in a series of theoretical studies for deep and recurrent networks in the large size limit, showing that ideal regimes for effective network performance are those initialized with such spectral attributes [29, 28, 7]. By construction, orthogonal matrices and their complex-valued counterparts, unitary matrices, are isometric operators and do not expand or contract space, which helps to mitigate the EVGP. A central challenge to train unitary RNNs is to ensure that parameter updates are restricted to the manifolds satisfying orthogonality constraints known as *Stiefel* manifolds (see review in [14]). This is an active area of optimization research, and several techniques have been used for orthogonal or unitary RNN training. In [2], the authors construct connectivity matrices with long products of rotation matrices leveraging fast Fourier transforms. In [33, 32, 10], the Cayley transform is used, which parametrizes weight matrices using skew-symmetric matrices that need to be inverted (cf. [6] for an RNN implementation directly using skew-symmetric matrices). Another approach uses Householder reflections [24]. Recent studies also adapt some of these methods to the quaternion domain [26]. The methods listed above have their advantages by either being fast, or memory efficient, but suffer from only parametrizing a subset of all orthogonal (unitary) matrices. A novel approach considering the group of unitary matrices as a Lie group and leveraging a parametrization via the exponential map applied to its Lie algebra, addresses this problem and currently outperforms the rest on many tasks [19]. Still, of all matrices with unit-norm eigenvalues, unitary matrices are only a small subset and remain limited in their expressivity since they are restricted to isometric transformations [11]. This is why orthogonal RNNs, while performing better than a conventional RNN or LSTM at some tasks (e.g. copy task [13], or sequential MNIST [18]), struggle at more complex tasks requiring computations across multiple timescales.

### 2.2 Non-normal connectivity

Any diagonalizable matrix $V$ can be expressed as $V = P\Theta P^{-1}$ where $P$'s columns are $V$'s eigenvectors and $\Theta$ is a diagonal matrix containing its eigenvalues. $V$ is said to be *normal* if its

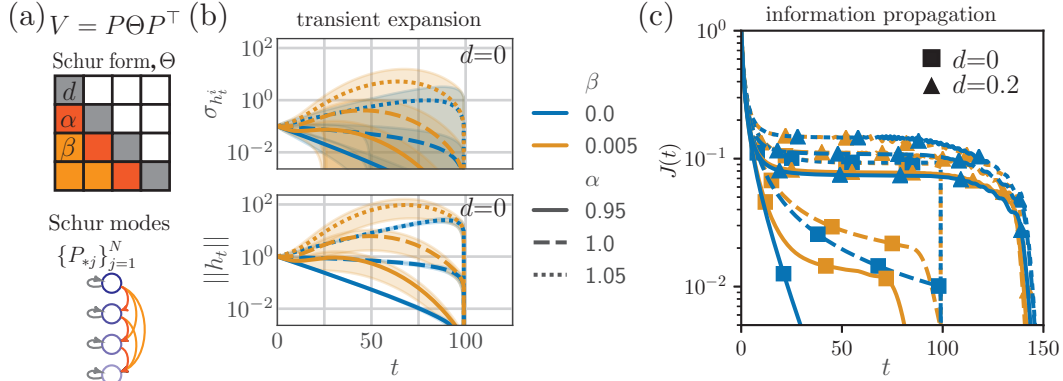

Figure 1: *Benefits of non-normal dynamics.* (a) The Schur decomposition provides the lower-triangular Schur form (top). A feed-forward interaction coupling among Schur modes underlies non-normal dynamics (bottom). (b) Lower triangle generates stronger transients. Trajectories of standard deviation across hidden units (top) and norm of hidden state vector (bottom) obtained from the dynamics of Eq. (1). Lines and shading are average and standard deviation, respectively, over $10^3$ initial conditions uniformly distributed on the unit hypersphere. Parameters: $d = 0$. (c) Fisher memory curves across $\alpha$ and $\beta$ (see legend in (b)) as computed by Eq. 7. Parameters: $d = 0$ (■), $d = 0.2$ (▲). $N = 100$ for (b) and (c).

eigenbasis is orthogonal and thus, $P^{-1} = P^\top$ and $V = P\Theta P^\top$. Orthogonal matrices are normal matrices with eigenvalues on the unit circle. When a matrix is *non-normal*, it is diagonalized with a non-orthogonal basis. However, it is still possible to express it using an orthogonal basis at the cost of adding (lower) triangular structure to $\Theta$. This is known as the *Schur* decomposition: for any matrix $V$, we have $V = P(\Lambda + T)P^\top$ with $P$ an orthogonal matrix, $\Lambda$ a diagonal matrix containing the eigenvalues, and $T$ a strictly lower-triangular matrix.[2] In short, $T$ contains the interactions between the orthogonal column vectors of $P$ (called *Schur modes*). $P$ and $T$ are obtained from orthogonalizing the non-orthogonal eigenbasis of $V$, and do not affect the eigenspectrum. As a recurrent connectivity matrix, $T$ represents purely feed-forward structure that produces strictly transient dynamics impossible to produce in normal (orthogonal) matrices. In other words, if a normal and non-normal matrix share exactly the same eigenspectrum, the iterative propagation of an input will be equivalent in the long-term, but can differ greatly in the short-term. We revisit this distinction in §3. This was exploited by [9, 12] to analyze the decomposition of the activity of recurrent networks (in continuous time) into a normal part responsible for slow fluctuations, and a non-normal part producing fast, transient ones. How this mechanism propagates information was studied in [8] for stochastic linear dynamics. The authors show analytically that non-normal dynamics can lead to extensive memory traces, as measured by the Fisher information of the distribution of hidden state trajectories parametrized by the input signal. To the best of our knowledge, an explicit demonstration and explanation of the benefits of non-normal dynamics for learning in RNNs is lacking, though see [25] for similar ideas used for initialization.

## 3 Non-normal matrices are more expressive and propagate information more robustly than orthogonal matrices

We now outline the role of non-normal connectivity we exploit for recurrent network parametrization. To provide mathematically-grounded intuition for the benefit it provides for learning, we first consider generic RNN dynamics,

$$h_{t+1} = \phi(V h_t + U x_{t+1} + b)$$
$$V = P\Theta P^\top, \ \Theta = \Lambda + T \tag{1}$$

where $h_t \in \mathbb{R}^N$ is the time-varying hidden state vector, $\phi$ is a nonlinear function, $x_t$ is the input sequence projected into the dynamics via matrix $U$, and $b$ is a bias (we omit the output for brevity).

$V$ is the matrix of recurrent weights, which in line with §2.2 we decompose into its lower-triangular Schur form $\Theta$ in Eq. (1), with $P$ orthogonal and $\Theta$ lower triangular. $\Theta$ has two parts: a (block) diagonal part $\Lambda$, and a strictly lower triangular part $T$.[3] The Schur decomposition maps the hard problem of controlling the directions of a non-orthogonal basis to the easier problem of specifying interactions between fixed orthogonal modes. It is important to highlight the fact that an orthonormalization of the eigenbasis is just a change in representation and thus has no effect on the spectrum of $V$, which still lies on the diagonal of $\Lambda$. The triangular part $T$ can thus be modified independently from the constraint (employed in orthogonal RNN approaches) that the spectrum have norms equal or near 1.

The ability to encode complex signals and then selectively recall past inputs is a basic requirement needed to solve many sequence-based tasks. Intuitively, the two features that allow systems to perform well in such tasks are:

1. High dimensional activity to better encode complex input.
2. Efficient signal propagation, to better learn long-term dependencies.

To illustrate how non-normal dynamics controlled by the entries in the lower triangle of $T$ contribute to these two features, we consider a simplified linear case where $\phi(h_t) = h_t$ and $\Theta$ is parametrized as follows and illustrated in Fig. 1(a)):

$$(\Theta)_{i,j} = d\delta_{i,j} + \alpha\delta_{i,j+1} + \beta \sum_{2 \leq k \leq i} \delta_{i,j+k}. \tag{2}$$

Here, diagonal entries are set to $d$, sub-diagonal entries to $\alpha$, and the remaining entries in the lower triangle to $\beta$. By varying $\alpha$ and $\beta$ we will show how the lower triangle in $T$ enhances expressivity and information propagation.

## 3.1 Non-normality drives expressive transients

RNNs can be made more expressive with stronger fluctuations of hidden state dynamics. The dependence of hidden state variance on the values of $\Theta$ was studied in depth in [12]. Here, we present experiments where the RNN parametrized by Eqs. (1), (2) exemplifies some of those results. We numerically compute a set of trajectories over a sampled ensemble of inputs with $x_t > 0$ for $t = 0$ and 0 otherwise. Without loss of generality we assume a form of $U$ and distribution of $x_0$ that leads to input-dependent initial conditions on the unit hypersphere in the space of $h_t$. For $\alpha = 0.95, 1.0, 1.05$, $\beta = 0, 0.005$ and $d = 0$, we see that trajectories of single units exhibit increasing large transients with increasing $\alpha$ and $\beta$, that abruptly end at $t = N$ (Fig. 1(b)). The latter is a result of the nilpotent property of a strictly triangular matrix: each iteration removes the top entries in each column until $\Theta^N = 0$. Computing ensemble statistics, we find that $\alpha$ contributes significantly to the strength of the exponential amplification, while $\beta$ structures the shape of the transient. This ability of $T$ to both exhibit amplification, and to control its shape, is what endows the Schur form $\Theta$ with expressivity (see §5.3 for empirical evidence in trained nnRNNs).

## 3.2 Non-normality allows for efficient information propagation

Propagation of information in a network requires feed-forward interactions. Perhaps the most simple example of a feed-forward structure is the local feed-forward chain (also called *delay-line* [8]), where each mode feeds its signal only to the next mode in the chain ($\alpha > 0$, $\beta = 0$, $d = 0$; see Fig. 1(a)). In this case, we denote $\Theta$ by $\Theta_{\text{delay}}$. As a consequence, signals feeding the first entry of $\Theta_{\text{delay}}$ propagate down the chain and are amplified or attenuated according to the values of these non-zero entries. Moreover, inputs from different time steps do not interact with each other thanks to this ordered propagation down the line. In contrast, the signal is not propagated across modes for dynamics given by a purely (block) diagonal $\Theta$. It instead simply decays within the mode into which it was injected on the timescale intrinsic to that mode, which can be much less than the $O(N)$ timescale of the chain.

To quantify the efficiency with which a RNN can store inputs, we follow and extend the approach of [8]. For a given scalar-valued input sequence, $x_t = s_t + \xi_t$, $t \in \mathbb{N}$, composed of signal $s_t$ and injected noise $\xi_t$, the noise ensemble induces the conditional distribution, $P(h_{:t}|s_{:t})$, over trajectories

of hidden states, $h_{:t}$, given the received input, $s_{:t}$, where $:t$ subscript is short hand for $(k : k \leq t)$. Taking the signal sequence $s_{:t}$ as a set of parameters of a model, and $P(h_{:t}|s_{:t})$ as this model's likelihood, the corresponding Fisher information matrix that captures how $P(h_{:t}|s_{:t})$ changes with the input $s_{:t}$ is,

$$\mathbf{J}_{k,l}(s_{:t}) = \left\langle -\frac{\partial^2}{\partial s_k \partial s_l} \log P(h_{:t}|s_{:t}) \right\rangle_{P(h_{:t}|s_{:t})} \qquad k, l \leq t. \tag{3}$$

The diagonal of this matrix, $J(t) := \mathbf{J}_{t,t}$ is called the Fisher memory curve (FMC) and has a simple interpretation: if a single signal $s_0$ is injected into the network at time 0, then $J(t)$ is the Fisher information that $h_t$ retains about this single signal.

In [8], the authors proved that the delay line $\Theta_{\text{delay}}$ achieves the highest possible values for the FMC when $k \leq N$: $J(k) = \alpha^k \frac{\alpha-1}{\alpha^{k+1}-1}$. However, we show (proof in SM§B) that any strictly lower-triangular matrix may approach the performance of a delay line:

**Proposition 1.** *Let $\Theta \in \mathbb{R}^{N \times N}$ be any strictly lower-triangular matrix with $\sqrt{\alpha}$ on the lower diagonal and let $T_{Gram} \in \mathbb{R}^{N \times N}$ be the triangular matrix associated with the Gram–Schmidt orthogonalization process of the columns of $\Theta$ (thus with only 1 on the diagonal). Then,*

$$J(k) \geq \frac{\alpha^k}{\sigma_{\max}^{2(N-1)}} \frac{\alpha-1}{\alpha^{k+1}-1}, \tag{4}$$

*where $\sigma_{\max}$ is the maximum singular value of $T_{Gram}$.*

Note that $\sigma_{\max} \geq 1$ and is equal to 1 for a delay line and close to 1 when $\Theta$ is close to a delay line. In Fig. 1(a) we present a class of matrices providing feed-forward interaction and compute the FMC of some matrices of this class in Fig. 1(c). The delay line from [8] with $\alpha > 1$ (shown to be optimal for $t \leq N$) retains the most Fisher information across time up to time step $N$, when the nilpotency of $\Theta$ erases all information. As expected from Prop.1, non-zero $\beta$, which endows the dynamics with expressivity (Fig. 1(b)), does not significantly degrade the information propagation of the delay line. Interestingly, the addition of diagonal terms ($d > 0$), i.e. $\Lambda$ non-zero, helps to maintain almost optimal values of the FMC for $t < N$, while extending the memory beyond $t = N$, and thus outperforming the delay line with regards to the area under the FMC (see Table 3 in the supplemental materials (SM)).

Together with the last section, these results demonstrate that non-normal dynamics, as parametrized through the entries in the lower triangle of $\Theta$, provide significant benefits to expressivity and information propagation. What remains to show is how these benefits translate into enhanced performance of our nnRNN on actual tasks.

### 3.3 Non-normal matrix spectra and gradient propagation

While eigenvalues control the exponential growth and decay of matrix iterates, the spectral norm of these iterates may behave differently [4]. This norm is dominated by the modulus of the largest singular value of the matrix, and can thus differ from the eigenvalues' moduli. This is a subtle difference influencing gradient growth rates, and is explicitly revealed by different spectral constraints on RNNs. For comparison, a singular value decomposition (SVD) is presented in [35] with the same motivation as our Schur-decomposition: to maintain expressivity, whilst controlling a sprectrum (both using regularization). First note that, while constraining the eigenspectrum to the unit circle, non-normality implies having the largest singular value (and thus the spectral norm of the Jacobian) greater than 1. Hence, our approach mitigates gradient vanishing, but not necessarily gradient explosion. In this case however, gradients explode polynomially in time rather than exponentially [27, 2]. We provide a theorem (proof in SM§C) to establish this for triangular matrices.

**Proposition 2.** *Let $A \in \mathbb{R}^{n \times n}$ be a matrix such that $A_{ii} = 1$, $A_{ij} = x$ for $i < j$, and $A_{ij} = 0$ otherwise. Then for all integer $t \geq 1$ and $j > i$, we have $(A^t)_{ij} = p_{j-i}^{(t)}(x)$ is polynomial in $x$ of degree at most $j - i$, where the coefficient of $x^0$ is zero and the coefficient of $x^l$ is $O(\binom{t}{l})$ for $l = 1, 2, \ldots, j - i$ (which is polynomial in $t$ of degree at most $l$).*

This reveals that gradient explosion in nnRNN with unit-norm eigenspectrum, if present, is polynomial and thus not as severe as the case where eigenvalues are larger than one (in which case the gradient

explosion is exponential). In §5.3, we illustrate that relaxing unit-norm requirements for eigenvalues using regularization allows the optimizer to find a task-dependent trade-off, thus balancing control over exponential vanishing and polynomial exploding gradients respectively. See also SM§F for gradient propagation measurements.

## 4 Implementing a non-normal RNN

The nnRNN is a standard RNN model where we parametrize the recurrent connectivity matrix $V$ using its real Schur decomposition[4] as in Eq. (1), yielding the form:

$$
V = P \left( \begin{bmatrix} \mathcal{R}_1 & 0 & \dots & 0 \\ 0 & \mathcal{R}_2 & \dots & 0 \\ \vdots & \vdots & \ddots & \vdots \\ 0 & 0 & \dots & \mathcal{R}_{N/2} \end{bmatrix} + \begin{bmatrix} 0 & 0 & \dots & 0 \\ t_{2,1} & 0 & \dots & 0 \\ \vdots & \vdots & \ddots & \vdots \\ t_{N,1} & t_{N,2} & \dots & 0 \end{bmatrix} \right) P^\top \tag{5}
$$

with

$$
\mathcal{R}_i(\gamma_i, \theta_i) \stackrel{\text{def}}{=} \gamma_i \begin{bmatrix} \cos\theta_i & -\sin\theta_i \\ \sin\theta_i & \cos\theta_i \end{bmatrix},
$$

where $P$ is constrained to be an $N \times N$ orthogonal matrix. Each parameter above (including entries in $P$) is subject to optimization, as well as specific constraints outlined below. We note that although this parametrization uses the Schur form, we never explicitly compute Schur decompositions, which would be expensive and has stability issues.[5] Note that Eq. 5 can express any matrix $V$ with a set of complex-conjugate pairs of eigenvalues.

During training, the orthogonal matrix $P$ is optimized using the expRNN algorithm [19], a Riemannian gradient descent-like algorithm operating inside the Stiefel manifold of orthogonal matrices. We note that other suitable orthogonality-preserving algorithms could be used here (see §2) but we found expRNN to be the fastest and most stable. Instead of rigidly enforcing that eigenvalues be of unit norm, we found relaxing this constraint to be helpful. We therefore allow $\gamma_i$ to be optimized but add a strong L2 regularization constraint $\delta \| \|1 - \gamma_i\| \|_2^2$ to encourage them to be to close to 1. The hyperparameter $\delta$ is tuned differently for each task (see SM§A) but remains high overall, indicating only mild departure from unit-norm eigenvalues. Both $\theta_i$ and $t_{ij}$ are freely optimized via automatic differentiation. The non-linearity we use is *modReLU*, as defined in [2, 10]. We initialize $P$ as in [19] using Henaff or Cayley initialization scheme [11], $\theta_i$ from a uniform distribution between 0 and $2\pi$, and $\gamma_i$'s are initialized at 1.

We reiterate that the set of orthogonal matrices is a subset of all the connectivity matrices covered by nnRNN, by setting all $\gamma$'s to 1, and $T = 0$. Consequently the connectivity matrix in nnRNN has more parameters than an orthogonal matrix: $N(N-1)/2$ for $T$, and $N/2$ $\gamma_i$'s, which in total gives roughly $N^2/2$ more parameters than orthogonal RNNs.

The forward pass of the nnRNN has the same complexity as that of a vanilla RNN, that is $O(Tn^2)$, for a hidden state of size $n$ and a sequence of length $T$. The backward pass is similarly $O(Tn^2)$ plus the update cost of $P$, in addition to a once-per-update cost of $O(n^3)$ to combine the Schur parametrization via matrix multiplication. Importantly, the nnRNN leverages any orthogonal/unitary optimizer for $P$ which have complexities ranging from $O(n \log n)$ to $O(n^3)$ at each update, with their own advantages and caveats (see §2.1). We chose the expRNN scheme, which is $O(n^3)$ in the worst case, but has fast run-time in practice.

## 5 Numerical experiments

In this section, we test the performance of our nnRNN on various sequential processing tasks. We have two goals:

1. Establish the nnRNN's ability to perform as well as orthogonal RNNs on tasks with pathologically long term dependencies: the copy task and the permuted sequential MNIST task.

2. Demonstrate improved performance over orthogonal RNNs on a more realistic task requiring ongoing computation and output: the Penn Tree Bank character-level benchmark.

We compare our nnRNN model to the following architectures: vanilla RNN (RNN), the orthogonally initialized RNN (RNN-orth) [11], the Efficient Unitary RNN (EURNN) [24], and the Exponential RNN (expRNN) [19]. Our goal is to establish performance for non-gated models, but we include LSTM [13] for reference. For comparison, models are separately matched in the number of hidden units and number of parameters. Every training run was tuned with a thorough optimization hyper-parameter search. Model training and task setup are detailed in SM§A.

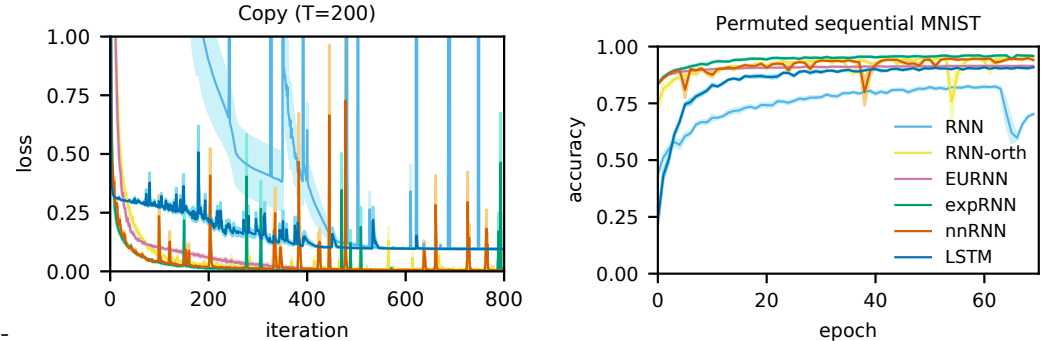

Figure 2: Holding the number $N$ of hidden units constant, model performance is plotted for the copy task (T=200, left; cross-entropy loss; $N \sim 128$) and for the permuted sequential MNIST task (right; accuracy; $N \sim 512$). Shading indicates one standard error of the mean.

## 5.1   Copy task & Permuted sequential MNIST

The copy task, introduced in [13], requires that a model reads a sequence of inputs, waits for some delay $T$ (here we use $T = 200$), and then outputs the same sequence. Fig. 2 shows the cross entropy of each tested with $N = 128$ hidden units. We see little difference if we match the number of parameters with $\sim 18.9K$ (see Fig. 4 in SM§A.2). For reference, a model that simply predicts a constant set of output tokens for every input sequence is expected to achieve a *baseline* loss of 0.095. As shown in [11], an orthogonal RNN is an optimal solution for the copy task. Indeed, the LSTM struggled to solve the task and RNN failed completely, unlike all orthogonal RNNs who learn to solve at very high performance very quickly. The proposed nnRNN matched the performance of orthogonal RNNs, as well as best training timescales.

Sequential MNIST [18] requires a model to classify an MNIST digit after reading the digit image one pixel at a time. The pixels are permuted in order to increase the time delay between inter-dependent pixels, making the task harder. Fig. 2 shows mean validation accuracy of each tested model with with $N = 512$ hidden units (see Fig. 4 in SM§A.2 for parameter match). As with the copy task, the nnRNN matches orthogonal RNNs in performance, whereas RNN and LSTM show lesser performances.

## 5.2   Penn Tree Bank (PTB) character-level prediction

Character level language modelling with the Penn Treebank Corpus (PTB) [22] consists of predicting the next character at each character in a sequence of text (see SM§A.3 for test accuracy). We compare the performance of different models on this task in Table 1 in terms of test mean bits per character (BPC), where lower BPC indicates better performance. We compare truncated backpropagation through time over 150 time steps and over 300 time steps.

In contrast to the copy and psMNIST tasks (see §5.1), the PTB task requires online computation across several inputs received in the past. Furthermore, it is a task that demands an output from the network at each time step, as opposed to a prompted one. These ingredients are not particularly well-suited for orthogonal transformations since it is not enough to simply keep inputs in memory or integrate input paths to a classification outcome, the network must transform past inputs to compute a probability distribution. Gated networks are well-suited for such tasks, and we could get an LSTM with $N = 1024$ hidden units to achieve $1.37 \pm 0.003$ BPC (see §6 for a discussion).

Importantly, without the use of gating mechanisms, our nnRNN outperformed all other models we tested. To our knowledge, it also surpasses all reported performances for other non-gated models

| Test Bit per Character (BPC) | | | | |
|---|---|---|---|---|
| | Fixed # params ($\sim$1.32M) | | Fixed # hidden units ($N = 1024$) | |
| Model | $T_{PTB} = 150$ | $T_{PTB} = 300$ | $T_{PTB} = 150$ | $T_{PTB} = 300$ |
| RNN | $2.89 \pm 0.002$ | $2.90 \pm 0.0016$ | $2.89 \pm 0.002$ | $2.90 \pm 0.002$ |
| RNN-orth | $1.62 \pm 0.004$ | $1.66 \pm 0.006$ | $1.62 \pm 0.004$ | $1.66 \pm 0.006$ |
| EURNN | $1.61 \pm 0.001$ | $1.62 \pm 0.001$ | $1.69 \pm 0.001$ | $1.68 \pm 0.001$ |
| expRNN | $1.49 \pm 0.008$ | $1.52 \pm 0.001$ | $1.51 \pm 0.005$ | $1.55 \pm 0.001$ |
| nnRNN | $\mathbf{1.47 \pm 0.003}$ | $\mathbf{1.49 \pm 0.002}$ | $\mathbf{1.47 \pm 0.003}$ | $\mathbf{1.49 \pm 0.002}$ |

Table 1: PTB test performance: Bit per Character (BPC), for sequence lengths $T_{PTB} = 150, 300$. Two comparisons across models shown: fixed number of parameters (left), and fixed number of hidden units (right). Error range indicates standard error of the mean.

of comparable size (see also [20]). While the performance gap to expRNN (the state-of-the-art orthogonal RNN) is modest for equal number of parameters and shorter time scale ($T_{PTB} = 150$), it appreciatively improves for $T_{PTB} = 300$. Where the nnRNN shines is for equal numbers of hidden units, where the performance gap to expRNN is much greater. This suggests two things (i) the nnRNN improves propagation of *meaningful* signals over longer time scales, and (ii) its connectivity structure provides superior expressivity for a fixed number of neurons, a desirable feature for efficient model deployment. In the next section, we explore the structure of trained nnRNN weights to illustrate that the mechanisms responsible for this performance gain are consistent with the arguments presented in §3.

## 5.3  Analysis of learned connectivity structure

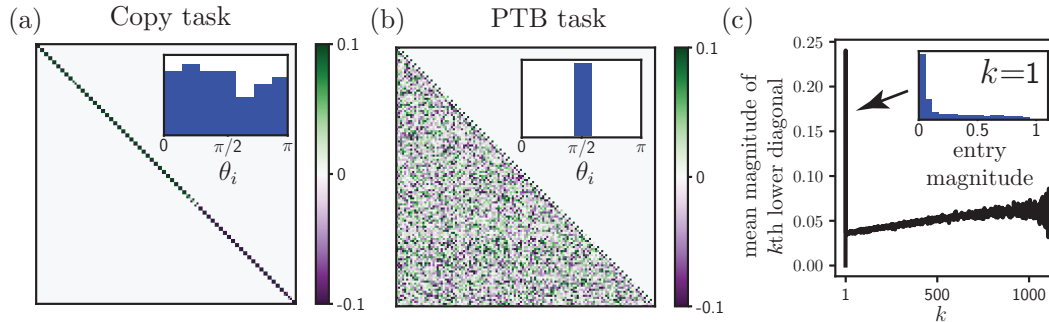

Figure 3: *Learned $\Theta$s show decomposition into $\Lambda$ and $T$.* Elements of learned $\Theta$ matrix entries for copy task (a) are concentrated on the diagonal, and distributed in the lower triangle for the PTB task (b). Insets in (a) and (b) show the distribution of eigenvalues angles $\theta_i$ (cf. Eq. 4). (c) The mean magnitude of entries along the $k$th sub-diagonal of the lower triangle in (b) shows both a delay-line and lower triangle component. Inset: the distribution of entry magnitudes along the delay line is bimodal from its two contributions: the cosine of uniformily distributed angles, and the relatively small, but significant pure delay line entries.

To validate the theoretical arguments in favor of non-normal dynamics presented in §3, we take a look at the connectivity structure that emerges from our training procedure (see §4). Fig. 3 (and 5 in SM§E) shows the triangular Schur form $\Theta = \Lambda + T$ of the recurrent connectivity matrix $V = P\Theta P^\top$, at the end of training. For the copy task, $\Theta$ is practically composed of $2 \times 2$ rotation blocks along its diagonal (i.e. $T = 0$). This indicates that the learned dynamics are normal, and orthogonal. In contrast, for the PTB task we find that the lower triangular part $T$ shows a lot of structure, indicating that non-normal transient dynamics are used to solve the prediction task. The distributions of elements of $T$ away from the diagonal highlights the nature of the tasks. The network distributes the angles roughly uniformly in the case of the copy task, consistent with the explicit optimal solution that involves such a distribution of rotations [11]. For the PTB task however, the angles strongly align, promoting the delay-line motif in $\Theta$, shown in §3 to be optimal for the information propagation useful for character prediction. This is more clearly demonstrated by the mean absolute value of entries away from the diagonal, shown in Fig. 3. The rest of the triangle also shows structure, consistent with our proof that the lower triangle and delay line can jointly contribute to information propagation.

In summary, these findings indicate that when tasks are well-suited for isometric transformations (e.g. storing things in memory for later recall) the nnRNN easily learns to eliminate non-normal dynamics and restricts itself to the set of orthogonal matrices. Moreover, it does so without any penalty on learning speed, as shown in Fig. 2. However, when tasks require online computations, non-normal dynamics come into play and enable transient activity to be shaped for computations.

Lastly, as already discussed in §3.3, the expressivity afforded by non-normality must come with a trade-off between maintaining the eigen and singular spectra "close" to the unit circle, balancing control over exponential vanishing and polynomial exploding gradients respectively. This fact remains true for any parametrization of non-normal matrices, including the SVD used in spectral RNN [35]. The nnRNN is naturally suited to target this balance by explicitly allowing regularization over normal and non-normal parts of a matrix, and enabling the optimizer to find that trade-off. This explains why we find that allowing eigenvalues to deviate slightly from the unit circle throughout training (regularization on $\gamma$), along with weight decay for the non-normal part, yields the best results with most stable training. Further evidence of this balancing mechanism is found in trained matrices (see Fig. 3). For the PTB task, non-normal structure emerges and the mean eigenvalue norm is balanced at $\bar{\gamma} \sim 0.958$. In contrast for the copy task, matrices remain normal and $\bar{\gamma} \sim 1$. See SM§F for additional experiments with fixed $\gamma$ further outlining their role in this trade-off.

## 6  Discussion

With the nnRNN, we showed that augmenting orthogonal recurrent connectivity matrices with non-normal terms increases the flexibility of a recurrent network. We compared the nnRNN's performance to several other recurrent models on distinct tasks; some that are well suited for orthogonal RNNs, and another that targets their limitations. We find that non-normal structure affords two distinct improvements for nnRNNs:

1. *Preservation of advantages from purely orthogonal RNNs* (long-term gradient propagation; fast learning on tasks involving long-term memory)

2. *Compared to orthogonal RNNs, increased expressivity on tasks requiring online computations thanks to transient dynamics.*

To better understand why this is, we derived analytical expressions that outline the role of non-normal dynamics that were corroborated by an analysis of nnRNN connectivity structure after training. Importantly, the nnRNN leverages existing optimization algorithms for orthogonal matrices with increased scope, all the while retaining learning speed.

The principal contribution of this paper is not to report major gains in performance as measured by tests, but rather to convincingly outline a promising novel direction for spectrally constrained RNNs. This spans the expressivity and ability to handle long-term dependencies of orthogonal RNNs on one hand, and completely unconstrained RNNs on the other. The nnRNN is a first step toward a trainable RNN parametrization where regularization over the eigenspectrum is readily available while conserving the flexibility of arbitrary eigenbases. This allows explicit control over quantities with direct impact on gradient propagation and expressivity, providing a promising RNN toolbox. Unlike the orthogonal RNNs present in our tests, which have benefited over the years from a series of algorithmic improvements, our nnRNN is basic in its implementation, and presents a number of areas for direct improvement. These include (i) using a complex-valued parametrization as in [2], (ii) exploring better initializations, and (iii) identifying helpful regularization schemes for the non-normal part. Beyond these, we should mention that the Schur decomposition presents implicit instabilities which can jeopardize training when eigenbases become degenerate (see SM§D). Simple perturbation schemes to prevent this should greatly improve performance.

Finally, we acknowledge that on a number of time-dependent tasks, gated recurrent networks such as the LSTM or the GRU [15] have clear advantages (see also [31] for a derivation of gated dynamics from first principles). Building on these, there is promising evidence that combining orthogonal connectivity with gates can greatly help learning [15]. This further motivates the development of spectrally constrained recurrent architectures to be combined with gating, thereby optimizing the efficiency of gradient propagation and expressivity with both explicit mechanisms, and implicit structure. Ongoing work in this direction is under way, leveraging our nnRNN findings.

**Acknowledgments**

We would like to thank Tim Cooijmans, Sarath Chandar, Jonathan Binas, Anirudh Goyal, César Laurent and Tianyu Li for useful discussions. YB ackowledges support from CIFAR, Microsoft and NSERC. MPT acknowledges IVADO support. GL is funded by an NSERC Discovery Grant (RGPIN-2018-04821), an FRQNT Young Investigator Startup Program (2019-NC-253251), and an FRQS Research Scholar Award, Junior 1 (LAJGU0401-253188).

## Footnotes

*Indicates first authors. Ordering determined by coin flip.

[2]When eigenvalues and eigenvectors are complex, $P$ is unitary and $P^\top$ corresponds to conjugate transposition. However for any real $V$, it is possible to find an orthogonal (real) $P$ with $\Lambda$ being block-diagonal with $2 \times 2$ blocks instead of complex-conjugate eigenvalues.

[3]We use the real Schur decomposition but a similar treatment can be derived for the complex case.

[4]See discussion for more details about complex-valued implementations.

[5]See SM§D for a discussion.

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
