[Supplementary Material]



**Supplemental Material for:**

Non-normal Recurrent Neural Network (nnRNN): learning long time dependencies while improving
expressivity with transient dynamics

# A    Task setup and training details

All code freely available at `https://github.com/nnRNN/nnRNN_release`.

## A.1    Copy task

For the copy task, networks are presented with an input sequence $x_t$ of length $10 + T_c$. For
$t = 1, \ldots, 10$, $x_t$ can take one of 8 distinct values $\{a_i\}_{i=1}^8$. For the following $T_c - 1$ time steps, $x_t$
takes the same value $a_9$. At $t = T_c$, a cue symbol $x_t = a_{10}$ prompts the model to recall the first 10
symbols and output them sequentially in the same order they were presented. Models are trained to
minimize the average cross entropy loss of symbol recalls. A model that simply predicts a constant
set of output tokens for every input sequence would achieve a *baseline* loss of $\frac{10 \log(8)}{T+20}$. All models
were trained using a mini batch size of 10. All non-gated models except "RNN" were initialized such
that the recurrent network was orthogonal. The non-normal RNN had it's orthogonal weight matrix
initialized as in expRNN with the log weights initialized using Henaff intialization. Importantly, all
non-gated models used the *modReLU* activation function for state-to-state transitions. This is critical
for the copy task since a nonlinearity makes the task very difficult to solve [VTKP17a] and *modReLU*
acts as identity at initialization. Fig. 4 (left) shows cross entropy loss for all models throughout
training when the number of parameters is held constant. Model and training hyperparameters are
summarized in Table 2.

| Model | hid | LR | LR orth | $\alpha$ | $\delta$ | $T$ decay | V init |
|---|---|---|---|---|---|---|---|
| nnRNN | 128 | 0.0005 | $10^{-6}$ | 0.99 | 0.0001 | $10^{-6}$ | Henaff |
| expRNN | 128 | 0.001 | 0.0001 | 0.99 | | | Henaff |
| expRNN | 176 | 0.001 | 0.0001 | 0.99 | | | Henaff |
| LSTM | 128 | 0.0005 | | 0.99 | | | Glorot Normal |
| LSTM | 63 | 0.001 | | 0.99 | | | Glorot Normal |
| RNN Orth | 128 | 0.0002 | | 0.99 | | | Random orth |
| EURNN | 128 | 0.001 | | 0.5 | | | |
| EURNN | 256 | 0.001 | | 0.5 | | | |
| RNN | 128 | 0.001 | | 0.9 | | | Glorot Normal |

Table 2: Hyperparameters for the copy task. Here, "hid" is hidden state size, "LR" is learning rate,
"LR orth" is the learning rate of the orthogonal transition matrix (its skew symmetric matrix), $\alpha$ is
the smoothing parameter of RMSprop, $\delta$ is as in equation 5, $T$ decay is the weight of the L2 penalty
applied on $T$ in equation 5, and "V init" is the initialization scheme for the state transition matrix.

## A.2    Sequential MNIST classification task

The sequential MNIST task [LJH15] measures the ability of an RNN to model complex long term
dependencies. In this task, each pixel is fed into the network one at a time, after which the network
must classify the digit. Permutation increases the difficulty of the problem by applying a fixed
permutation to the sequence of the pixels, which creates longer term dependencies between the pixels.
We train this task for all networks using mini batch sizes of 100. All non-gated networks except "RNN"
were initialized with orthogonal recurrent weight matrices using Cayley initialization[HWY18].
The non-normal RNN has it's orthogonal weight matrix initialized as in [LCMR19] with the log
weights initialized using Cayley initialization. Fig. 4 (right) shows validation accuracy for all
models throughout training when the number of parameters is held constant. Model and training
hyperparameters are summarized in Table 3.

## A.3    Penn Tree Bank character prediction task

The Penn Tree Bank character prediction task is that of predicting the next character in a text corpus
at every character position, given all previous text. We trained all models sequentially on the entire

Figure 4: Holding the number of parameters constant, model performance is plotted for the copy task (T=200, left; cross-entropy loss; 18.9K parameters) and for the permuted sequential MNIST task (right; accuracy; 269K parameters). Shading indicates one standard error of the mean.

| Model | hid | LR | LR orth | $\alpha$ | $\delta$ | $T$ decay | V init |
|---|---|---|---|---|---|---|---|
| nnRNN | 512 | 0.00015 | $1.5 * 10^{-5}$ | 0.99 | 0.15 | 0.0001 | Cayley |
| expRNN | 512 | 0.0005 | $5 * 10^{-5}$ | 0.99 | | | Cayley |
| expRNN | 722 | 0.0005 | $5 * 10^{-5}$ | 0.99 | | | Cayley |
| LSTM | 512 | 0.0005 | | 0.9 | | | Glorot Normal |
| LSTM | 257 | 0.0005 | | 0.9 | | | Glorot Normal |
| RNN Orth | 512 | $5 * 10^{-5}$ | | 0.99 | | | Random orth |
| EURNN | 512 | 0.0001 | | 0.9 | | | |
| EURNN | 1024 | 0.0001 | | 0.9 | | | |
| RNN | 512 | 0.0001 | | 0.9 | | | Glorot Normal |

Table 3: Hyperparameters for the permuted sequential mnist task. Here, "hid" is hidden state size, "LR" is learning rate, "LR orth" is the learning rate of the orthogonal transition matrix (its skew symmetric matrix), $\alpha$ is the smoothing parameter of RMSprop, $\delta$ is as in equation 5, $T$ decay is the weight of the L2 penalty applied on $T$ in equation 5, and "V init" is the initialization scheme for the state transition matrix.

corpus, splitting it into sequences of length 150 or 300 for truncated backpropagation through time. Consequently, the initial hidden state for a sequence is the last hidden state produced from its preceding sequence. All models were trained for 100 epochs with a mini batch size of 128. Following training, for each model, the state which yielded the best performance on the validation data was evaluated on the test data. Table 2 reports the same performance for the same model states as in Table 1 in the main text but presents test accuracy instead of BPC. Model and training hyperparameters are summarized in Table 5.

| | Test Accuracy | | | |
|---|---|---|---|---|
| | Fixed # params ($\sim$1.32M) | | Fixed # hidden units ($N = 1024$) | |
| Model | $T_{PTB} = 150$ | $T_{PTB} = 300$ | $T_{PTB} = 150$ | $T_{PTB} = 300$ |
| RNN | $40.01 \pm 0.026$ | $39.97 \pm 0.025$ | $40.01 \pm 0.026$ | $39.97 \pm 0.025$ |
| RNN-orth | $66.29 \pm 0.07$ | $65.53 \pm 0.09$ | $66.29 \pm 0.07$ | $65.53 \pm 0.09$ |
| EURNN | $65.68 \pm 0.002$ | $65.55 \pm 0.002$ | $64.01 \pm 0.002$ | $64.20 \pm 0.003$ |
| expRNN | $68.07 \pm 0.15$ | $67.58 \pm 0.04$ | $67.51 \pm 0.11$ | $66.89 \pm 0.024$ |
| nnRNN | $\mathbf{68.78 \pm 0.0006}$ | $\mathbf{68.52 \pm 0.0004}$ | $\mathbf{68.78 \pm 0.0006}$ | $\mathbf{68.52 \pm 0.0004}$ |

Table 4: PTB test performance: Test Accuracy, for sequence lengths $T_{PTB} = 150, 300$. Two comparisons across models shown: fixed number of parameters (left), and fixed number of hidden units (right). Error range indicates standard error of the mean.

| Model | hid | LR | LR orth | $\alpha$ | $\delta$ | $T$ decay | V init |
|---|---|---|---|---|---|---|---|
| **Length 150** | | | | | | | |
| nnRNN | 1024 | 0.0008 | $8*10^{-5}$ | 0.9 | 1 | 0.0001 | Cayley |
| expRNN | 1024 | 0.005 | 0.0001 | 0.9 | | | Cayley |
| expRNN | 1386 | 0.005 | 0.0001 | 0.9 | | | Cayley |
| LSTM | 1024 | 0.008 | | 0.9 | | | Glorot Normal |
| LSTM | 475 | 0.001 | | 0.99 | | | Glorot Normal |
| RNN Orth | 1024 | 0.0001 | | 0.9 | | | Random orth |
| EURNN | 1024 | 0.001 | | 0.9 | | | |
| EURNN | 2048 | 0.001 | | 0.9 | | | |
| RNN | 1024 | $10^{-5}$ | | 0.9 | | | Glorot Normal |
| **Length 300** | | | | | | | |
| nnRNN | 1024 | 0.0008 | $6*10^{-5}$ | 0.9 | 0.0001 | 0.0001 | Cayley |
| expRNN | 1024 | 0.005 | 0.0001 | 0.9 | | | Cayley |
| expRNN | 1386 | 0.005 | 0.0001 | 0.9 | | | Cayley |
| LSTM | 1024 | 0.008 | | 0.9 | | | Glorot Normal |
| LSTM | 475 | 0.003 | | 0.9 | | | Glorot Normal |
| RNN Orth | 1024 | 0.0001 | | 0.9 | | | Cayley |
| EURNN | 1024 | 0.001 | | 0.9 | | | |
| EURNN | 2048 | 0.001 | | 0.9 | | | |
| RNN | 1024 | $1*10^{-5}$ | | 0.9 | | | Glorot Normal |

Table 5: Hyperparameters for the Penn Tree Bank task (at 150 and 300 time step truncation for gradient backpropagation). Here, "hid" is hidden state size, "LR" is learning rate, "LR orth" is the learning rate of the orthogonal transition matrix (its skew symmetric matrix), $\alpha$ is the smoothing parameter of RMSprop, $\delta$ is as in equation 5, $T$ decay is the weight of the L2 penalty applied on $T$ in equation 5, and "V init" is the initialization scheme for the state transition matrix.

### A.4 Hyperparameter search

For all models with a state transition matrix that is initialized as orthogonal (nnRNN, expRNN, RNN-orth), three orthogonal initialization schemes were tested: (1) random, (2) Cayley, and (3) Henaff. Random initialization is achieved by sampling a random matrix whose QR decomposition yields an orthogonal matrix with positive determinant 1 and then mapping this orthogonal matrix via a matrix logarithm to the skew symmetric parameter matrix used in expRNN. Cayley and Henaff initializations initialize this skew symmetric matrix as described in [LCMR19]. The vanilla RNN is also tested with a Glorot Normal initialization, with the model then referred to as simply "RNN".

For training, learning rates were searched between 0.01 and 0.0001 in increments of 0.0001, 0.0002 or 10×; the learning rate for the orthogonal matrix was always kept near 10× lower; and RMSprop was used as the optimizer with smoothing parameter $\alpha$ as 0.5, 0.9, or 0.99. In equation 5, $\delta$ was searched in 0, 0.0001, 0.001, 0.01, 0.1, 0.15, 1.0, 10; the L2 decay on the strictly upper triangular part of the transition matrix $T$ was searched in 0, $10^{-6}$, $10^{-5}$, $10^{-4}$.

## B  Fisher Memory Curves for strictly lower-triangular matrices

Let, $\Theta$ be a strictly lower triangular matrix such that $[\Theta]_{i+1,i} = \sqrt{\alpha}$ for $1 \leq i \leq N-1$ and $A$ be the associated lower triangular Gram-Schmidt orthogonalization matrix. We have that,

$$\Theta = DA \tag{6}$$

where $D$ is the delay line, $D_{i+1,i} = \sqrt{\alpha}$ and $A_{i,i} = 1$ for $1 \leq i \leq N$. Let us recall the expression of $J(k)$ for independent Gaussian noise derived by [GHS08b, Eq. 3],

$$J(k) = U^T(\Theta^k)^\top C_n^{-1} \Theta^k U \,, \quad \text{where} \quad C_n = \epsilon \sum_{k=0}^{\infty} \Theta^k (\Theta^k)^\top \,, \tag{7}$$

451 and $U = [1, 0, \ldots, 0]$ is the source. We have that for any vector $u$,

$$u^\top C_n u = \epsilon \sum_{k=0}^{\infty} ((D^k)^\top u)^\top A A^T ((D^k)^\top u) \tag{8}$$

$$= \epsilon \sum_{k=0}^{N-1} ((D^k)^\top u)^\top A A^T ((D^k)^\top u) \tag{9}$$

$$\leq \epsilon \sigma_{\max}^{2(N-1)}(A) \sum_{k=0}^{N-1} u^\top D^k (D^k)^\top u \tag{10}$$

452 where for the first equality we used the fact that $\Theta$ is nilpotent and for the last inequality the fact that
453 $\sigma_{\max}(A) \geq 1$. Recall that for two symmetric matrices we define: $A \succeq B$ if and only if $B - A$ is
454 positive semidefinite. By definition we have,

$$C_n \preceq \epsilon \sigma_{\max}^{2(N-1)}(A) \sum_{k=0}^{\infty} D^k (D^k)^\top = \epsilon \sigma_{\max}^2(A) \left( \mathrm{diag}(1, \tfrac{1-\alpha^2}{1-\alpha}, \ldots, \tfrac{1-\alpha^N}{1-\alpha}) \right) \tag{11}$$

455 where the last equality is due to $[D^k(D^k)^\top]_{i,j} = \alpha^k$ if $i = j \geq k+1$ and $0$ otherwise. Thus
456 using [Lax07, Theorem 2 P. 146] we can take the inverse to get,

$$C_n^{-1} \succeq \frac{1}{\epsilon \sigma_{\max}^{2(N-1)}(A)} \left( \mathrm{diag}(1, \tfrac{1-\alpha^2}{1-\alpha}, \ldots, \tfrac{1-\alpha^N}{1-\alpha}) \right)^{-1} = \frac{1}{\epsilon \sigma_{\max}^{2(N-1)}(A)} \mathrm{diag}(1, \tfrac{1-\alpha}{1-\alpha^2}, \ldots, \tfrac{1-\alpha}{1-\alpha^N})$$

457 Finally, using that $\Theta^k U = [\underbrace{0, \ldots, 0}_{k}, \sqrt{\alpha}^{-k}, *, \ldots, *]$, we have that for $0 \leq k \leq N-1$,

$$J(k) = U^T (\Theta^k)^\top C_n^{-1} \Theta^k U \tag{12}$$

$$\geq \frac{1}{\epsilon \sigma_{\max}^{2(N-1)}(A)} \alpha^k \frac{\alpha - 1}{\alpha^{k+1} - 1} . \tag{13}$$

| $\alpha$ | $\beta$ | $d$ | $J_{\text{tot}} = \sum_{t=0}^{\infty} J(t)$ |
|---|---|---|---|
| 0.95 | 0.0 | 0.0 | 3.03 |
| 1.00 | 0.0 | 0.0 | 5.19 |
| 1.05 | 0.0 | 0.0 | 12.1 |
| 0.95 | 0.005 | 0.0 | 3.18 |
| 1.00 | 0.005 | 0.0 | 5.30 |
| 1.05 | 0.005 | 0.0 | 12.1 |
| 0.95 | 0.0 | 0.2 | 12.0 |
| 1.00 | 0.0 | 0.2 | 16.2 |
| 1.05 | 0.0 | 0.2 | 20.5 |
| 0.95 | 0.005 | 0.2 | 12.1 |
| 1.00 | 0.005 | 0.2 | 16.3 |
| 1.05 | 0.005 | 0.2 | 20.4 |

Table 6: Fisher memory curve performance: Shown is the sum of the FMC for the models considered in section 3.

458

# C   Numerical instablities of the Schur decomposition

460 The Schur decomposition is computed via multiple iterations of the QR algorithm. The QR algorithm
461 is known to be *backward stable*, which gives accurate answers as long as the eigenvalues of the matrix
462 at hand are well-conditioned, as is explained in [ABB+99].

463 Eigenvalue-sensitivity is measured by the angle formed between the left and right eigenvectors of
464 the same eigenvalues. Normal matrices have coinciding left and right eigenvectors but non-normal

465 matrices do not, and thus certain non-normal matrices such as the Grcar matrix have very high
466 eigenvalue-sensitivity, and thus gives rise to inaccuracies in the Schur decomposition.

467 This motivates training the connectivity matrix in the Schur decomposition directly instead of applying
468 the Schur decomposition in a separate step.

## D   Learned connectivity structure on psMNIST

Figure 5: *.Learned Θ on psMNIST task*. Inset: angles $\theta_i$ distribution of block diagonal rotations. (c.f.
Eq.4).

470 For completeness, let us take a look at the Schur matrix after training on psMNIST in Fig. 5. We
471 can see that the distribution of learned angles in the rotation blocks is rather flat, and thus is very
472 different from the distribution learned in the PTB task, as can be seen in Fig. 3. The flatness in
473 distribution comes somewhat close to the flatness of the learned angle distribution in the copy task.
474 In other words, the angle distribution in the PTB task is highly structured, while in the Copy task and
475 psMNIST task, it seems to be close to uniform.

476 Furthermore, we can also observe that the connectivity structure learned in the lower triangle is
477 significantly weaker in the psMNIST task than in the PTB task, while not being completely absent as
478 in the copy task.

479 Thus it seems that we can spot a spectrum of connectivity structure:

480 • the Copy task, with no connectivity structure in the lower triangle, close to uniform angle
481   distribution and the absence of a delay line, on the one end.
482 • the PTB task, with a lot of connectivity structure in the lower triangle, a very narrow angle
483   distribution and the presence of a delay line, on the other end.

484 For the psMNIST task, it appears that we are located somewhere in the middle of that spectrum.