[Reviews · NeurIPS 2019]

Reviewer 1



The authors introduce a new method to enhance the expressiveness of orthogonal RNN via the Schur decomposition, based on previous work by Ganguli et al. Orthogonal/ unitary RNN surpass gated RNN (LSTM, GRU) in tasks that involve long-term dependencies, but are limited in computational expressiveness due to the unitary constraint. Using the Schur decomposition for non-normal matrices, the authors partly overcome this limitation, with the off-diagonal elements of the Schur matrix enabling information propagation and interactions between the different modes. Strengths: In my mind this is a strong paper that makes important contributions to the field, further advancing a promising concept for solving the vanishing/exploding grad problem in RNN. As the authors stated, their work is strongly rooted in previous work by Ganguli et al., but through further simulations and analyses they demonstrate how these ideas could boost unitary RNN. Weaknesses: Given that the authors argue convincingly that the Schur decomposition should give much larger expressiveness, it is somewhat surprising that the performance gap between nnRNN and LSTM (0.12) is still twice as large as the one between nnRNN and expRNN (0.06). This seems to indicate that nnRNN are not really that much further ahead than the best-performing orthogonal RNN when compared to LSTM. How do the authors explain this, is this just due to the gating mechanism that nnRNN lack? Minor: - Isn’t the result in Fig. 1b completely expected, given that an orthogonal RNN lacks the off-diagonal elements (which could boost hidden states) in the Schur form?

Reviewer 2



Originally: The ideas presented in the paper are original. Quality: The theoretical and experimental analysis seems to be properly conducted. However, as the non-normal approach is compared to the performance of the expRNN and LSTM, comments should be added, how the asymptotic (in number of units or parameters) runtimes of the different approaches compare. Especially the runtime of optimizing P (line 187-189) should be clarified. Clarity: The paper is clearly written. Significance: Solving the proplem of vanishing and exploiding gradients is of large importance and at the same time a very difficult endeavor. Extending the orthogonal RNN approach and fixing its problems is one way to tackle this problem. The paper contributes novel theoretical ideas towards fixing this problems. The experimental improvement seems rather small and thus the immediate significance is probably limited. However, this work might be a step towards solving the vanishing and exploiding gradient problem. _____________________ After reading the author feedback and reviewer discussion I increase my score to 6. I think overall the paper has strong contributions and valuable insights, though tackling a hard problem. However, I ask the authors to state the asymptotic runtime of the proposed algorithms in the final version of the manuscript. I think it would be very bad to hide the O(n^3) runtime from the reader.

Reviewer 3



This paper proposes nnRNN, which is the non-normal matrix relaxation to improve the expressivity of RNNs with unit eigenvalue constraints. Previous studies use the unitary or orthogonal matrix constraints to prevent the exploding and vanishing gradient problems in RNNs, but these constraints deteriorate the performance in empirical tasks due to the strong constraints. This paper relaxes the unitary matrix (which is a normal matrix that has eigenvalues on the unit circle) constraints as allowing the non-normal matrices that have the eigenvalues on a unit circle. In addition, this paper theoretically and empirically shows how the non-normal relaxation affects the dynamics of RNNs. This paper is well written, and the motivation of non-normality is explained thoroughly. The main results are original and interesting since there are few studies that focus on whether the recurrent matrix is a normal matrix or non-normal matrix. However, I tend to vote to reject due to the following issues. 1. There is no guarantee that nnRNN solves the exploding gradient problem. The URNN paper [ASB16] shows that URNN can solve the exploding gradient problem due to the spectral norm (the largest singular value) of one rather than due to eigenvalues on a unit circle. The singular values of non-normal matrices can be larger than one even when they have eigenvalues of one. Since the spectral norm is larger than or equal to spectral radius (the largest absolute value of eigenvalues), the gradient vanishing might be prevented by nnRNN. However, nnRNN might not prevent gradient from exploding. As a simple example, we consider the case of a linear RNN h_{t+1}=Vh_{t} where V=[1,2; 0,1]. In this case, V is a non-normal matrix and its eigenvalues are 1 while its spectral norm is larger than 1. The Jacobian of dh_{T}/dh_{0} can be \prod_{i=0}^{T-1} dh_{i+1}/dh_{i}=V^{T}= [1,2T;0,1], and the norm of Jacobian increases according to T. Therefore, there is no guarantee that non-normal matrices can solve the exploding gradient problem even if they can alleviate exploding gradient. In fact, Fig. 1 shows that multiplying the non-normal matrix increases the norm of the hidden vector. This implies that the chain rule can make the gradient explode by using non-normal matrices. 2. The experimental condition might be unfair. In experiments, the eigenvalues of nnRNN are optimized as gamma while the eigenvalues of other methods are constrained to be one. Due to this difference, the contribution of non-normal matrix relaxation is not clear; the improvements in nnRNN might be caused by the relaxation of eigenvalues. -------------- I have read the author feedback. My questions are mostly solved, and interesting and important discussions will be provided. Therefore, I raise the overall score.

[Author Response · NeurIPS 2019]

We highlight the delightful situation where reviews considerably enhanced the impact of our results without any structural changes to the story, and thank the reviewers for it. Importantly, R3's remarks led to an expanded interpretation of non-normality in RNNs which, combined to our proposed nnRNN model, we believe are of important significance to the understanding of RNN gradients. An updated version of our paper incorporates the points below.

**R1: Q**"*Add more explanation and insight on why the gap to LSTM performance is still larger than that to expRNN.*" **A:** We believe this is due to the task design which suits gated networks very well (LSTM performs poorly on the other two tasks tested). As indicated in the main text, our goal is to perfect parametrization of recurrent connectivity, which can then be combined to gated architecture (ongoing and future work).

**R2: Q**"*[Clarify] how the asymptotic (in number of units or parameters) runtimes of the different approaches compare. Especially the runtime of optimizing P.*" **A:** The forward pass of the nnRNN has the same complexity as that of a vanilla RNN, that is $O(Tn^2)$, for a hidden state of size $n$ and a sequence of length $T$. The backward pass is similarly $O(Tn^2)$ plus the update cost of P, in addition to a once-per-update cost of $O(n^3)$ to combine the Schur parametrization via matrix multiplication. Importantly, the nnRNN leverages any orthogonal/unitary optimizer fof P which have complexities ranging from $O(n \log n)$ to $O(n^3)$ at each update, with their own advantages and caveats (see related work section in main text). We chose the expRNN scheme which is $O(n^3)$ in the worst case, but has fast run-time in practice.

**R3: Q**"*There is no guarantee that nnRNN solves the exploding gradient problem. [...] comparison [of] nnRNN with spectral RNN might provide important insights. I recommend adding the theoretical or empirical analysis of the exploding gradient problems.*" **A:** We thank R3 for pointing us to the *spectral RNN* (Zhang, Lei, and Dhillon, ICML 2018), which presents an SVD decomposition with the same motivation as our to Schur-decomposition: to maintain expressivity, whilst controlling a sprectrum (both using regularization). R3 astutely remarks that "*[the relationship between these methods could] reveal whether we should constrain the eigenvalues or singular values of recurrent connection matrices*". We strongly believe that this distinction is important and not well understood by the community, and that the changes described herein add clarity to this question, in addition to strengthening the existing message of our paper.

As pointed out, constraining the eigenspectrum to the unit circle mitigates gradient vanishing, but not necessarily gradient explosion, as singular values can still be greater than one (and so too the spectral norm of Jacobians). In this case however, gradients explode polynomially in time rather than exponentially (Pascanu et al. (2013), Arjovsky et al. (2015)). We provide a theorem to establish this for triangular matrices. **Theorem:** Let $A \in R^{n \times n}$ be a matrix such that $A_{ii} = 1$, $A_{ij} = x$ for $i < j$, and $A_{ij} = 0$ otherwise. Then for all $d \geq 1$ and $j > i$, we have $(A^d)_{ij} = p^{(d)}_{j-i}(x)$ is polynomial in $x$ of degree at most $j - i$, where the coefficient of $x^0$ is zero and the coefficient of $x^l$ is $O(\binom{d}{l})$ for $l = 1, 2, \ldots, j - i$. (proof by induction will be presented in appendix) This reveals that: **(1) Gradient explosion in nnRNN, if present, is not as severe as if eigenvalues were larger than one**. As shown below, training of nnRNN with eigenvalues strictly on the unit circle may be successful, albeit somewhat unstable, but still more expressive than orthogonal RNNs. The figure shows that gradients for nnRNN on PTB task, with eigenvalues clamped or regularized, behave nicely during backpropagation and throughout training. **(2) For a matrix with eigenvalues on the unit circle, non-normality necessarily implies a largest singular value greater than one. Thus, the expressivity afforded by non-normality must come with a trade-off between maintaining the eigen and singular spectra "close" to the unit circle, balancing control over exponential vanishing and polynomial exploding gradients respectively**. This fact remains true for any parametrization of non-normal matrices, including the SVD used in spectral RNN. The

nnRNN is naturally suited to target this balance by explicitly allowing regularization over normal and non-normal parts of a matrix, and enabling the optimizer to find that trade-off. This explains why, in the main text, we find that allowing eigenvalues to deviate slightly from the unit circle throughout training (regularization on $\gamma$), along with weight decay for the non-normal part, yields the best results with most stable training. Further evidence of this balancing mechanism is found in trained matrices (see Fig 3 in main text). For the PTB task, non-normal structure emerges and the mean eigenvalue norm is balanced at $\bar{\gamma} \sim 0.958$. In contrast for the copy task, matrices remain normal and $\bar{\gamma} \sim 1$. Our Schur approach complements that of the SVD approach in additional ways: by adding the freedom to distribute eigenvalues on the unit circle (which we showed was exploited by the learning), and by adding interpretability as interaction between modes (which revealed task-intuitive differences in the solutions).

**R3:**"*Since optimization of the gamma obscures the cause of improvements, you should compare nnRNN with gamma=1 to other methods.*" **A:** Thanks to the results above, we now know that optimization of $\gamma$ plays an intricate role in allowing the expressivity afforded by non-normal connectivity structure while conserving good gradient propagation. Nevertheless, we acknowledge that it confounds the expressive role of non-normality. To elucidate this, we train the nnRNN on PTB by clamping $\gamma$ at 1, and at 0.958 (the mean values found

| Model | $T_{PTB} = 150$ | $T_{PTB} = 300$ |
|---|---|---|
| nnRNN-$\gamma = 1$ | $1.46 \pm 0.005$* | $1.49 \pm 0.022$** |
| nnRNN-$\gamma = 0.958$ | $1.47 \pm 0.005$ | $1.49 \pm 0.008$ |

by the optimizer in the unclamped case) respectively. Results complement those of Table 1 in the main text ($\sim$1.32M params, $N = 1024$ units). As expected for $\gamma = 1$, some run did not converge (asterisks indicate number out of 5) as the emergence of non-normal structure pushes singular values above one. Despite this, on runs that did converge we found the best performance out of all methods (including regularized $\gamma$ nnRNN), strongly indicating that non-normality does indeed provide more expressivity. For $\gamma$ clamped at 0.958 the performance was virtually identical to that of nnRNN with regularized $\gamma$, indicating non-normal connectivity learning appears robust and independent of $\gamma$ learning. Exploration of novel ways to promote the balance between polynomial explosion and exponential vanishing is promising future work.



[Meta-Review · NeurIPS 2019]

The paper contributes in a significant way to show how RNN able to deal with vanishing/exploding gradients can be obtained not only by enforcing recurrent weight matrices to be normal or orthogonal, but also by exploiting a larger set of matrices obtained by Schur matrix decomposition. Although the proposed contribution cannot be readily extended to gated units and the computational effort is increased because of the Schur matrix decomposition, the paper provides new insights that can be the basis for improved algorithms. The rebuttal helped to solve most of the issues raised by the reviewers.